# Deep Learning-Based Prediction Model for Gait Recovery after a Spinal Cord Injury

**DOI:** 10.3390/diagnostics14060579

**Published:** 2024-03-08

**Authors:** Hyun-Joon Yoo, Kwang-Sig Lee, Bummo Koo, Chan-Woo Yong, Chae-Won Kim

**Affiliations:** 1Korea University Research Institute for Medical Bigdata Science, Korea University, Seoul 02841, Republic of Korea; 2AI Center, Korea University Anam Hospital, Korea University College of Medicine, Seoul 02841, Republic of Korea; 3School of Health and Environmental Science, Korea University College of Health Science, Seoul 02841, Republic of Korea; bummo0624@naver.com (B.K.); fogbow0313@naver.com (C.-W.Y.); kcherry@korea.ac.kr (C.-W.K.)

**Keywords:** spinal cord injury, deep learning, recurrent neural network, linear regression, Ridge, Lasso, prediction, somatosensory evoked potential

## Abstract

Predicting gait recovery after a spinal cord injury (SCI) during an acute rehabilitation phase is important for planning rehabilitation strategies. However, few studies have been conducted on this topic to date. In this study, we developed a deep learning-based prediction model for gait recovery after SCI upon discharge from an acute rehabilitation facility. Data were collected from 405 patients with acute SCI admitted to the acute rehabilitation facility of Korea University Anam Hospital between June 2008 and December 2022. The dependent variable was Functional Ambulation Category at the time of discharge (FAC-DC). Seventy-one independent variables were selected from the existing literature: basic information, International Standards for Neurological Classification of SCI scores, neurogenic bladders, initial FAC, and somatosensory-evoked potentials of the lower extremity. Recurrent neural network (RNN), linear regression (LR), Ridge, and Lasso methods were compared for FAC-DC prediction in terms of the root-mean-squared error (RMSE). RNN variable importance, which is the RMSE gap between a complete RNN model and an RNN model excluding a certain variable, was used to evaluate the contribution of this variable. Based on the results of this study, the performance of the RNN was far better than that of LR, Ridge, and Lasso. The respective RMSEs were 0.3738, 2.2831, 1.3161, and 1.0246 for all the participants; 0.3727, 1.7176, 1.3914, and 1.3524 for those with trauma; and 0.3728, 1.7516, 1.1012, and 0.8889 for those without trauma. In terms of RNN variable importance, lower-extremity motor strength (right and left ankle dorsiflexors, right knee extensors, and left long toe extensors) and the neurological level of injury were ranked among the top five across the boards. Therefore, initial FAC was the seventh, third, and ninth most important predictor for all participants, those with trauma, and those without trauma, respectively. In conclusion, this study developed a deep learning-based prediction model with excellent performance for gait recovery after SCI at the time of discharge from an acute rehabilitation facility. This study also demonstrated the strength of deep learning as an explainable artificial intelligence method for identifying the most important predictors.

## 1. Introduction

A spinal cord injury (SCI) can be defined as “damage to any part of the spinal cord or nerves at the end of the spinal canal” [1]. This is a common factor in permanent alterations in body functions below the damaged position. This results from damage to the spinal cord or surrounding bones and tissues. The degree of disability is determined by the degree and location of an injury. Damage to the upper spinal cord can lead to impairment of the limbs and body. Damage to the lower spinal cord can result in leg and lower body impairment. Complete recovery is difficult once an injury has occurred [2]. Global incidence, prevalence, and years of life lost due to disability from an SCI were 0.9 million cases, 20.6 million cases, and 6.2 million years, respectively, in 2019 [3]. The mean age of patients with SCI increased from 28.3 years in 1978 to 37.1 years in 2008 in the United States. The disease burden is higher in the elderly because of health issues. As life expectancy increases, the risk factors of falls, osteoporosis, and spinal stenosis increase [3], and medical costs are also high. The medical cost for a quadriplegic patient aged 25 years is estimated to be USD 5.16 million in the United States. Finally, the socioeconomic burden is significant for patients and their families alike [4]. Gait dysfunction is a crucial sequela, and recovery of walking ability is a high priority for patients with SCI in terms of both physical independence and self-esteem [5].

In this context, predicting gait recovery after an SCI during the acute rehabilitation phase is of paramount importance in designing rehabilitation strategies. For individuals who are expected to recover sufficiently with independent gait function, rehabilitation approaches mainly focus on using restorative techniques, such as endurance training, balance training, and lower-extremity strengthening, to promote neuroplasticity and enhance independent gait. In contrast, for individuals with limited potential for neurorecovery, there is more emphasis on compensatory techniques, such as wheelchair mobility or bed transfer [6]. Additionally, prediction of gait function is important when establishing a discharge plan. Based on their functional independence, patients are transferred to a subacute inpatient rehabilitation facility or discharged and allowed to return home. However, previous studies on the prediction of walking ability after an SCI have some limitations. First, most studies have focused on the effects of individual factors that influence walking recovery. For example, a review article stated that the American Spinal Injury Association Impairment Scale (AIS) score at admission, age, etiology, sex, evoked potentials, and magnetic resonance imaging findings, such as the presence and size of hemorrhages, were found to be important prognostic factors for walking ability [7]. However, few studies have considered the interaction between each factor and optimized modeling by considering the confounding factors between each variable. Second, previous prediction-modeling studies focused on final gait function when neurological recovery reached a plateau. In a representative study by Van Middendorp et al., a clinical prediction rule was developed based on age and clinical neurological parameters, such as motor and sensory scores, to predict the probability of walking independently after a one-year traumatic SCI [8]. Other studies combining neurophysiological data have also attempted to predict gait function at six months or one year after SCI prognosis and recovery among patients with ischemic and traumatic SCIs [9,10].

Artificial intelligence (AI) has surpassed conventional statistical approaches in terms of performance measures for predicting various disease conditions. However, limited AI studies have been conducted on this topic. Only one recent review summarized the recent applications of machine learning in the clinical diagnosis, prognostication, and management of acute traumatic SCIs [11]. Thirteen original studies were selected according to the following eligibility criteria: (1) having a dependent variable consisting of functional ability after an acute SCI; (2) reporting interventions using a decision tree, tree ensemble (random forest and boosting), support vector machine, artificial neural networks, and/or convolutional neural networks; (3) reporting outcomes of accuracy and F1 score (harmonic mean of sensitivity and specificity); (4) having a publication year of 2010 or later; and (5) being published in the English language. Among the 13 AI articles, only 3 studies were conducted to predict ambulatory function. One study compared the performances of unsupervised machine learning and logistic regression in the prognosis of walking ability [12]. Another study used a regression tree model to predict functional outcomes after traumatic SCI. [13]. Only one deep learning study has attempted to quantify radiographic characteristics by using a convolutional neural network to stratify neurological prognosis [14].

Recurrent neural networks (RNNs) have been widely adopted in sequential data research, including for prediction [15]. However, to the best of our knowledge, no studies are available on the application of an RNN in this direction. The primary purpose of this study was to develop a deep learning-based prediction model for gait recovery after SCI at discharge from an acute rehabilitation facility. Furthermore, we explored the important variables that were highly correlated with gait prognosis using a deep learning approach.

## 2. Methods

### 2.1. Participants

Data were obtained from electronic medical records of 427 patients with acute SCI who were admitted to the acute rehabilitation facility of Korea University Anam Hospital between June 2008 and December 2022. Demographic data and study variables were retrospectively reviewed. From the data set, we extracted data of all adult patients (≥18 years) with acute SCI, including cauda equina syndrome. Finally, patients with complete medical information on the study variables were included. Children and adolescents with SCI and those with missing data were excluded. Finally, data from 405 participants were analyzed. This study was approved by the Institutional Review Board of Korea University Anam Hospital (2022AN0473). Requirement for informed consent was waived by the institutional review board.

### 2.2. Variables

The dependent variable was the Functional Ambulation Category at the time of discharge (FAC-DC), a clinical indicator of six-level walking ability in terms of independence from physical support (where “0” indicates “no independence” and “5” indicates “complete independence”) [16]. Seventy-one independent variables were selected from the existing literature [16,17,18,19] (Table 1): (1) basic information (8 predictors), i.e., age, elderly (≥65 years) (yes vs. no), gender (female), period of acute care (days), period of rehabilitation (days), SCI etiology (traumatic vs. non-traumatic), cardiovascular disease (yes vs. no), and diabetes mellitus (yes vs. no); (2) all the neurological information entered on the International Standards for Neurological Classification of SCI Worksheet (60 predictors), that is, damaged part (C, T, L, and cauda equina), the neurological level of injury (cervical, thoracic, lumbar, or cauda equina), AIS (A, B, C, D, and E), injury completeness (yes vs. no), 4 motor scores (left or right upper or lower extremity), 4 sensory scores (light touch or pin prick, left or right), 10 lower-extremity strength values of each myotome, 18 light touch scores of each dermatome, 18 pin prick scores of each dermatome, voluntary anal contraction (yes vs. no), and deep anal pressure; and (3) three other predictors, including neurogenic bladder (yes vs. no), initial FAC, and somatosensory evoked potential (SSEP) of lower extremity (normal, prolonged/shallow, and not available). All independent variables were recorded within the first three days after admission to the rehabilitation ward.

### 2.3. Analysis

The RNN (long short-term memory), linear regression (LR), Ridge, and Lasso were compared for the prediction of FAC-DC. The RNN has one long short-term memory layer with tangent hyperbolic as an activation function, sigmoid as a recurrent activation function, a dropout of 0.0, a recurrent dropout of 0.0, and a time step of 1. Other hyper-parameters were Adam as an optimizer, an epoch number of 100, and a batch size of 30. Data from 405 participants were divided into training and validation sets in a 75:25 ratio (304 vs. 101 observations). A criterion used for validating the models trained was the root-mean-squared error (RMSE), that is, the root for the average of the squares of errors among 101 observations. RNN variable importance, which is the RMSE gap between a complete RNN model and an RNN model excluding a certain variable, was used to evaluate the contribution of variables. For example, we assume that the RNN variable importance of the ankle dorsiflexor, Rt, is 0.0044. This indicates that the inclusion of the variable ankle dorsiflexion, Rt, in the RNN reduces the RMSE of the model by 0.0044. A random split and analysis were repeated ten times, and the average was used for external validation. Different seed numbers were used for different runs; however, the default parameters remained the same throughout the random splits and analyses. Regarding statistical power, the RMSE over the standard deviation can be considered equivalent to the inverse of the power in the case of the RNN for the prediction of a continuous variable. Python 3.52 (Centrum voor Wiskunde en Informatica, Amsterdam, The Netherlands) was used for analysis in August 2023.

## 3. Results

### 3.1. Descriptive Statistics

The medical records of 405 adult patients with acute SCI were reviewed retrospectively. Descriptive statistics for continuous variables and their correlations with the FAC-DC are presented in Table 1. The proportions of those with categorical characteristics were as follows: elderly status (49%, 197), female sex (36%, 146), trauma (43%, 175), hypertension (48%, 195), diabetes (27%, 109), voluntary anal contraction (78%, 315), deep anal pressure (85%, 346), neurogenic bladder (57%, 229), and abnormal somatosensory evoked potential of lower extremity (79%, 169 for prolonged or shallow and 153 for no response). Likewise, 39, 21, 68, 275, and 2 participants had AIS levels of A, B, C, D, and E, respectively. Neurological levels C5 and C4 corresponded to the greatest number of participants (76 and 73, respectively). In the correlation analysis, initial FAC and lower-extremity motor strength were highly correlated with FAC-DC; the correlations of FAC-DC with ankle dorsiflexor, right knee extensor, left long toe extensor, and initial FAC were 0.62, 0.63, 0.62, 0.62, and 0.74, respectively. Therefore, age and the period of acute care before rehabilitation were negatively correlated with FAC-DC, suggesting that older age and delayed rehabilitation negatively impact prognosis.

### 3.2. Model Peformance

Based on these results, we presumed that the initial FAC and lower-extremity strength significantly affected gait recovery at discharge. Therefore, four scenarios were considered for a more detailed analysis: (1) initial FAC included and four motor scores (upper or lower extremity, left or right) included; (2) initial FAC excluded and four motor scores included; (3) initial FAC included and two motor scores (upper extremity, left or right) excluded; and (4) initial FAC included and four motor scores excluded. Therefore, we tried to analyze the impact of initial FAC by comparing scenarios (1) and (2) and of lower-extremity strength by comparing scenarios (3) and (4).

The performance of the RNN exceeded that of LR, Ridge, and Lasso across all scenarios. For example, the respective RMSEs averaged over ten runs in the first scenario (main scenario) were 0.3738, 2.2831, 1.3161, and 1.0246 for all the participants; 0.3727, 1.7176, 1.3914, and 1.3524 for those with trauma; and 0.3728, 1.7516, 1.1012, and 0.8889 for those without trauma (Table 2). The presence or absence of independent variables (initial FAC and lower-extremity strength) did not have a significant effect on model performance.

### 3.3. Vairable Importance

The top 20 significant predictors of the FAC-DC are presented in Table 3 and Figure 1, Figure 2 and Figure 3. In terms of RNN variable importance, ankle dorsiflexors, right knee extensors, left long toe extensors, and neurological level of injury were ranked among the top five across the board in all the patient groups. Initial FAC was the seventh, third, and ninth most important predictor for all participants, those with trauma, and those without trauma, respectively. Demographic features such as age, period of acute care before rehabilitation, and duration of rehabilitation were also selected as important variables. Additionally, the results of the electrophysiologic study (SSEP) also significantly impacted gait prognosis in both the traumatic and non-traumatic SCI groups.

## 4. Discussion

Predicting gait recovery after an SCI during the acute rehabilitation phase is of paramount importance in designing rehabilitation strategies. However, to date, few studies have been conducted on this topic. In this study, we developed a deep learning-based prediction model with excellent performance for gait recovery after an SCI at the time of discharge from an acute rehabilitation facility. This study also demonstrated the strength of deep learning as an explainable artificial intelligence method for identifying the most important predictors. The performance of the RNN was far better than that of LR, Ridge, and Lasso, and the respective RMSEs were 0.3738, 2.2831, 1.3161, and 1.0246 for all the participants; 0.3727, 1.7176, 1.3914, and 1.3524 for those with trauma; and 0.3728, 1.7516, 1.1012, and 0.8889 for those without trauma. In terms of RNN variable importance, lower-extremity motor strength (both ankle dorsiflexors, right knee extensor, and left long toe extensor) and neurological level of injury were among the top five predictors for all the groups. The initial FAC ranked seventh, third, and ninth for all the participants, those with trauma, and those without trauma, respectively.

The accurate prediction of prognosis for a certain disease or trauma has been one of the main goals of rehabilitation medicine. For this reason, many studies have been conducted to evaluate prognostic factors of various functional outcomes such as AIS, Barthel Index, Spinal Cord Independence Measure, and respiratory dysfunction [20,21,22]. However, there has been a lack of research on developing prediction models that integrate such predictors. Only one study developed a user-friendly clinical prediction rule that can predict the long-term probability of walking independently on the basis of age and clinical parameters such as motor and sensory scores [8]. One of the main reasons for this dearth is that the traditional statistical methods have some limitations with respect to dealing with larger numbers of predictors that are rather complex and have nonlinear relationships within datasets. To overcome such disadvantages, AI has also been recently applied in the predictive modeling of various outcomes (e.g., quality of life, duration of opioid prescription and duration of intensive care unit stay or mortality) after SCI [23,24,25]. According to a recent review published in 2022, only three published articles report the use of a machine learning approach or image-based deep learning analysis for gait prediction [11]. More recently, another study was published that established machine learning models for predicting spinal cord independence measures using features present at the time of rehabilitation admission. The cited study was a retrospective study with clinical data on 210 patients with SCI. They used RMSE and mean absolute error to assess model performance and concluded that the random forest model was superior in the training group, while a stacked generalization model better predicted spinal cord independence measure in the testing group [26]. However, to the best of our knowledge, no study has been conducted using RNN to predict functional status at discharge from an acute rehabilitation hospital for patients with SCI.

This study has several clinical implications. First, this study focused on deep learning as a strong foundation for a decision support system for gait recovery after SCI. Based on the results of a recent review [11], different machine learning methods are appropriate for different tasks in the clinical diagnosis, prognostication, and management of acute traumatic SCI, namely, the ensemble tree in the case of numeric data (accuracy 81.1%), and the convolutional neural network in the case of image data. However, to the best of our knowledge, no studies have been conducted on the application of an RNN in this direction. As described above, the performance of the RNN surpassed that of LR, Ridge, and Lasso by a large margin across the board. To the best of our knowledge, this study is the first to be conducted in this manner, and further studies are required on this topic. Second, this study validated the existing literature on the important roles of lower-extremity motor scores, neurological level of injury, and initial FAC as major predictors of gait recovery after SCI [27,28,29,30,31,32,33]. As explained above, the variable importance values of lower-extremity motor strength and neurological level of injury were among the top five for all groups. Additionally, the ranking of the initial FAC was seventh, third, and ninth for all the participants and those with and without trauma, respectively. Third, the rehabilitation period was found to have a strong association with gait recovery, emphasizing the importance of rehabilitation. Fourth, the effects of light touch and pin-prick sensory scores on gait recovery were not significant at discharge in this study, even though these effects were significant one year after an SCI in a previous study [8]. One possible explanation for this discrepancy is that this study focused on gait recovery at discharge, whereas the previous study focused on gait recovery during the chronic phase. Further examination of this topic is needed for the effective management of gait recovery after SCI.

Despite its clinical implications, this study has some limitations. First, it did not consider multimodal deep learning. Combining deep learning models for image, text, and numeric data is expected to significantly improve model performance. Secondly, the scope of this study did not include explainable reinforcement learning [34]. This cutting-edge approach has been popular in finance because of its realistic assumptions and excellent performance [35]. Its popularity has spread to areas such as diagnostic automation and treatment recommendations in healthcare [36]. However, little analysis has been conducted on explainable reinforcement learning, warranting further investigation. Finally, the proposed model was not externally validated. We concluded that the sample size was not large enough to construct a validation set. Instead, we used internal validation with data splitting. Further study with external validation set is needed to advance this prediction model.

## 5. Conclusions

Despite these limitations, this study developed a deep learning-based prediction model with excellent performance for gait recovery after an SCI at the time of discharge from an acute rehabilitation facility. Additionally, this study demonstrates the strength of deep learning as an explainable artificial intelligence method for identifying the most important predictors. By precisely predicting gait function, this study will aid in personalizing rehabilitative care for patients with acute SCI.

## Figures and Tables

**Figure 1 diagnostics-14-00579-f001:**
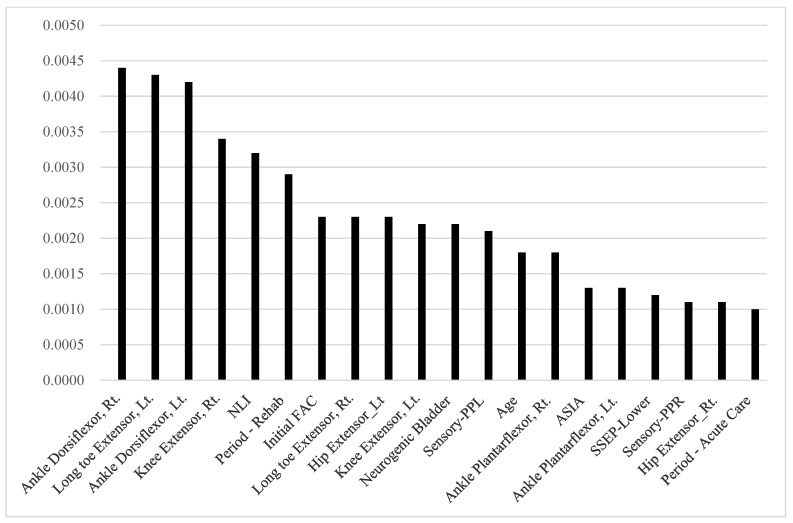
Recurrent neural network variable importance: 405 participants.

**Figure 2 diagnostics-14-00579-f002:**
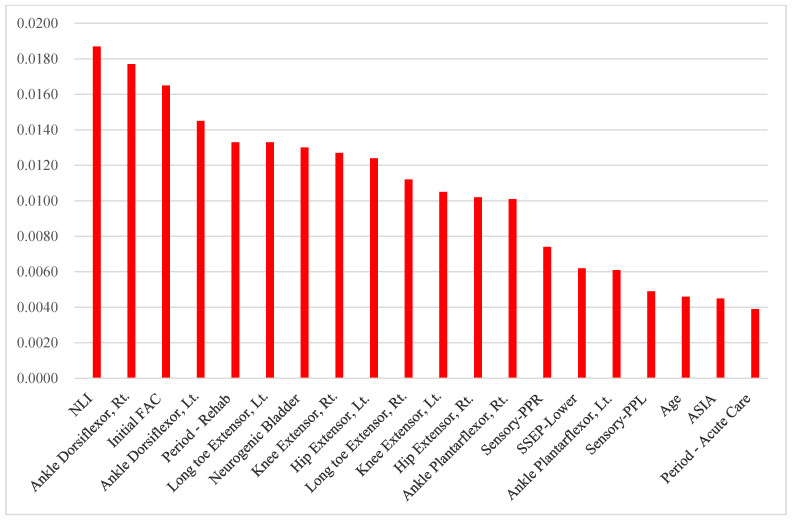
Recurrent neural network variable importance: 175 participants (Trauma: Yes).

**Figure 3 diagnostics-14-00579-f003:**
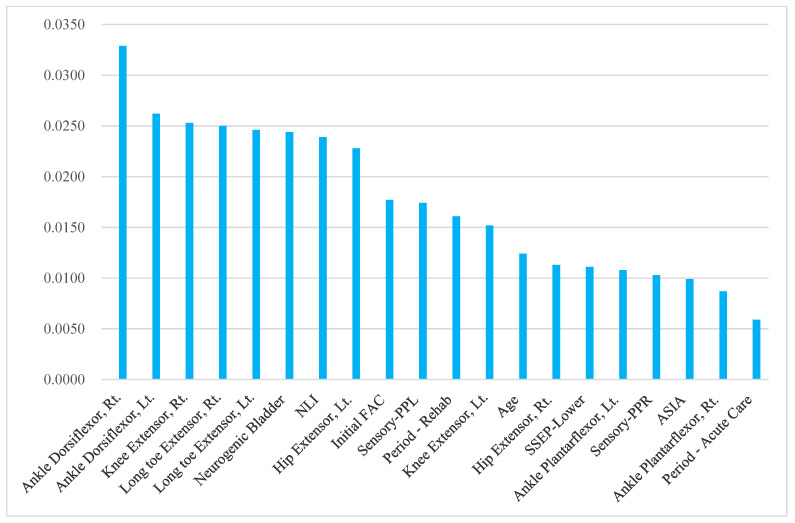
Recurrent neural network variable importance: 230 participants (Trauma: No).

**Table 1 diagnostics-14-00579-t001:** Descriptive statistics and FAC-DC predictor correlation. Legend: The correlations of FAC_DC with right ankle dorsiflexor, left ankle dorsiflexor, right knee extensor, left long toe extensor, and initial FAC were 0.62, 0.63, 0.62, 0.62, and 0.74, respectively. Neurological levels of injury C5 and C4 had the greatest numbers of participants, i.e., 76 and 73.

Variable	Min	25%	50%	75%	Max	SD	Correlation
FAC_DC	0	1	2	4	5	1.7	1.00
Age	38	52	64	74	98	15.8	−0.03
Period—Acute Care	0	17	38	133	10,642	945.2	−0.09
Period—Rehab	0	20	27	33	349	34.0	−0.13
Motor-UER	0	17	24	25	25	6.5	0.30
Motor-UEL	0	17	23	25	25	6.7	0.32
Motor-LER	0	7	19	22	25	8.8	0.64
Motor-LEL	0	8	18	22	25	8.7	0.64
Sensory-LTR	2	32	44	55	56	12.0	0.38
Sensory-LTL	2	32	45	55	56	12.5	0.36
Sensory-PPR	0	33	45	56	56	13.1	0.40
Sensory-PPL	0	32	45	56	56	13.2	0.37
Hip flexor,_Rt.	0	2	3	4	5	1.7	0.65
Knee extensor,_Rt.	0	2	4	5	5	1.8	0.62
Ankle dorsiflexor,_Rt.	0	1	4	4	5	1.9	0.62
Long toe extensor,_Rt.	0	1	3	4	5	1.8	0.61
Ankle plantarflexor,_Rt.	0	2	4	5	5	1.8	0.61
Hip flexor,_Lt.	0	2	3	4	5	1.7	0.66
Knee extensor,_Lt.	0	2	4	5	5	1.8	0.63
Ankle dorsiflexor,_Lt.	0	1	4	4	5	1.9	0.63
Long toe extensor,_Lt.	0	1	4	4	5	1.8	0.62
Ankle plantarflexor,_Lt.	0	2	4	5	5	1.8	0.61
LT_L1_Rt	0	1	1	2	2	0.6	0.44
LT_L2_Rt	0	1	1	2	2	0.6	0.44
LT_L3_Rt	0	1	1	2	2	0.6	0.43
LT_L4_Rt	0	1	1	2	2	0.6	0.45
LT_L5_Rt	0	1	1	2	2	0.6	0.42
LT_S1_Rt	0	1	1	2	2	0.6	0.44
LT_S2_Rt	0	1	1	2	2	0.7	0.45
LT_S3_Rt	0	1	1	2	2	0.7	0.45
LT_S4-5_Rt	0	1	1	2	2	0.7	0.43
LT_L1_Lt	0	1	1	2	2	0.7	0.41
LT_L2_Lt	0	1	1	2	2	0.7	0.41
LT_L3_Lt	0	1	1	2	2	0.7	0.40
LT_L4_Lt	0	1	1	2	2	0.7	0.42
LT_L5_Lt	0	1	1	2	2	0.7	0.39
LT_S1_Lt	0	1	1	2	2	0.7	0.42
LT_S2_Lt	0	1	1	2	2	0.7	0.42
LT_S3_Lt	0	1	1	2	2	0.7	0.40
LT_S4-5_Lt	0	1	1	2	2	0.7	0.38
PP_L1_Rt	0	1	1	2	2	0.7	0.43
PP_L2_Rt	0	1	1	2	2	0.7	0.45
PP_L3_Rt	0	1	1	2	2	0.7	0.46
PP_L4_Rt	0	1	1	2	2	0.7	0.46
PP_L5_Rt	0	1	1	2	2	0.7	0.45
PP_S1_Rt	0	1	1	2	2	0.7	0.47
PP_S2_Rt	0	1	1	2	2	0.7	0.47
PP_S3_Rt	0	1	1	2	2	0.7	0.46
PP_S4-5_Rt	0	1	1	2	2	0.7	0.44
PP_L1_Lt	0	1	1	2	2	0.7	0.42
PP_L2_Lt	0	1	1	2	2	0.7	0.42
PP_L3_Lt	0	1	1	2	2	0.7	0.45
PP_L4_Lt	0	1	1	2	2	0.7	0.45
PP_L5_Lt	0	1	1	2	2	0.7	0.43
PP_S1_Lt	0	1	1	2	2	0.7	0.45
PP_S2_Lt	0	1	1	2	2	0.7	0.45
PP_S3_Lt	0	1	1	2	2	0.7	0.44
PP_S4-5_Lt	0	1	1	2	2	0.7	0.42
Initial FAC	0	0	1	2	5	1.4	0.74

Note: LEL/LER—Lower Extremity, Left or Right; Lt—Left; LT—Light Touch; PP—Pin Prick; Rt—Right; UEL/UER—Upper Extremity, Left or Right.

**Table 2 diagnostics-14-00579-t002:** Model Performance: Root-mean-squared error averaged over 10 Runs. Legend: The performance of the recurrent neural network far exceeded that of LR, Ridge, and Lasso across the board. For example, their respective root-mean-squared errors in the first scenario (main scenario) were: 0.3738, 2.2831, 1.3161, and 1.0246 for all participants; 0.3727, 1.7176, 1.3914, and 1.3524 for those with trauma; and 0.3728, 1.7516, 1.1012, and 0.8889 for those with non-traumatic SCI.

		All Participants (*n* = 405)	Participants with Traumatic SCI (*n* = 175)	Participants with Non-Traumatic SCI (*n* = 230)
** *FAC Included* **	** *Motor Included* **			
**RNN**		**0.3738**	**0.3727**	**0.3728**
**Linear Regression**		**2.3831**	**1.7176**	**1.7516**
**Ridge**		**1.3161**	**1.3914**	**1.1012**
**Lasso**		**1.0246**	**1.3524**	**0.8889**
*FAC Excluded*	*Motor Included*			
RNN		0.3727	0.3727	0.3732
Linear Regression		2.2952	2.1961	1.5994
Ridge		1.4387	1.6006	1.2638
Lasso		1.1107	1.6058	1.1576
*FAC Included*	*Motor Upper* *excluded*			
RNN		0.3732	0.3727	0.3731
Linear Regression		1.9811	1.6266	1.9296
Ridge		1.2287	1.3526	1.1840
Lasso		1.0387	1.2526	0.9177
*FAC Included*	*Motor Excluded*			
RNN		0.3727	0.3727	0.3728
Linear Regression		1.2074	1.6294	1.8930
Ridge		1.1506	1.3383	1.1870
Lasso		1.0568	1.2258	0.9178

Note: **Bold** indicates **Main Scenarios**; FAC—Functional Ambulation Category; RNN—Recurrent Neural Network.

**Table 3 diagnostics-14-00579-t003:** Recurrent neural network variable importance. Legend: In terms of recurrent neural network variable importance, right ankle dorsiflexor, left ankle dorsiflexor, right knee extensor, left long toe extensor, and the neurological level of injury ranked among the top 5 across the board. Indeed, the initial functional ambulation category was the 7th, 3rd, and 9th most important predictor for all participants, those with trauma, and those without trauma, respectively.

Variable	All Participants (*n* = 405)		Participants with Traumatic SCI (*n* = 175)		Participants with Non-Traumatic SCI (*n* = 230)	
	VI	Ranking	VI	Ranking	VI	Ranking
Age	0.0018	13	0.0046	18	0.0124	13
Period—Acute Care	0.0010	20	0.0039	20	0.0059	20
Period—Rehab	0.0029	6	0.0133	5	0.0161	11
AIS	0.0013	15	0.0045	19	0.0099	18
NLI	0.0032	5	0.0187	1	0.0239	7
Sensory-PPR	0.0011	18	0.0074	14	0.0103	17
Sensory-PPL	0.0021	12	0.0049	17	0.0174	10
Hip flexor, Rt.	0.0011	18	0.0102	12	0.0113	14
Knee extensor, Rt.	0.0034	4	0.0127	8	0.0253	3
Ankle dorsiflexor, Rt.	0.0044	1	0.0177	2	0.0329	1
Long toe extensor, Rt.	0.0023	8	0.0112	10	0.0250	4
Ankle plantarflexor, Rt.	0.0018	13	0.0101	13	0.0087	19
Hip flexor, Lt.	0.0023	8	0.0124	9	0.0228	8
Knee extensor, Lt.	0.0022	10	0.0105	11	0.0152	12
Ankle dorsiflexor, Lt.	0.0042	3	0.0145	4	0.0262	2
Long toe extensor, Lt.	0.0043	2	0.0133	5	0.0246	5
Ankle plantarflexor, Lt.	0.0013	15	0.0061	16	0.0108	16
Neurogenic Bladder	0.0022	10	0.0130	7	0.0244	6
Initial FAC	0.0023	7	0.0165	3	0.0177	9
SSEP-Lower	0.0012	17	0.0062	15	0.0111	15

Note: AIS—American Spinal Injury Association Impairment Scale; NLI—Neurological Level of Injury; PPR/PPL—Pin Prick, Left/Right; FAC—Functional Ambulation Category; SSEP—Lower Somatosensory Evoked Potential of Lower Extremity; VI—Variable Importance.

## Data Availability

The raw data supporting the conclusions of this article will be made available by the authors on request.

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
