# Peer review of "Deep Learning-Based Prediction Model for Gait Recovery after a Spinal Cord Injury"

_diagnostics, 2024, doi:10.3390/diagnostics14060579_

Round 1

Reviewer 1 Report

Comments and Suggestions for Authors

introduction section need more clarify about previous studies in this field especially review studies.

method section need sample size calculation, inclusion and exclusion criteria of subject selection. in addition power of the your study must be add to the method section.

results section need explain your outcome measures separately.

Reviewer 2 Report

Comments and Suggestions for Authors

I have reviewed the paper "Deep Learning-Based Prediction Model for Gait Recovery", I consider that the paper does not contain the minimum necessary information regarding the use of deep learning models, the observations would be the following:

Section: 2.3. Analysis

Observation: There are hyperparameters that are very relevant for neural network training and should be mentioned in the publication such as: optimizer, learning rate, batch size, number of epochs, etc.

Action/Suggestion: Include the relevant hyperparameters used during the training of the neural network, if many experiments were performed with different combinations of these hyperparameters, then mention at least the ones used with the neural network that obtained the best results.

Section: 2.3. Analysis

Observation: When artificial neural networks are used, three data sets are required. The training set is used for the model to learn. The validation set is used to adjust the hyperparameters and control the training process. Finally, to verify that the model is viable for implementation or use, a test data set is needed, which contains data not used during training. If all three data sets are not used, it is not possible to state that the model will be able to predict correctly on new data.

Action/Suggestion: Justify why the three data sets were not used and their implications and add that in the publication.

Section: 2.3. Analysis

Observation: No information or graph about the recurrent neural network model or architecture used can be found. This is important since the novelty of the article is the use of a recurrent neural network model.

Action/Suggestion: Please add descriptive information about the recurrent neural network model used (number of layers, nodes, activation functions, etc.), and if possible, complement such information with a graph or figure of the architecture of the proposed model.

Section: 3. Results

Observation: The paper does not show comparisons with similar proposals.

Action/Suggestion: It is recommended that the authors add a comparison with similar proposals in narrative form or in a table, or justify why it was not done.

Reviewer 3 Report

Comments and Suggestions for Authors

1.       Terms like “L4_rt”, although clear need to be defined the first time they are mentioned, both in the abstract as well as in the main text.

2.       Need clearer impact statement. More elaboration is needed in the Introduction on how the proposed model will impact the design of rehabilitation strategies.   

3.       Was cross-fold validation performed on the models?

4.       In methods, more details are needed when describing the variable importance. In addition to the results, the method section should describe what variables were excluded for estimating variable importance, and why.

5.       Results were wrapped up in just 1 paragraph. Many details were skipped or not described completely.

6.       What was the reason for using ‘motor scores’ as criteria for developing the 4 scenarios for AI analysis in the results section.

7.       Table tiles and the values in the rows are not aligned. Please center align the values as the title of each column.

8.       Table 3: explanation needed for “yes Ranking”, “no ranking” and “Trauma VI” in the result section.

Comments on the Quality of English Language

 The manuscript needs major English language edits. Many statements like “The degree of its disability is determined by its degree and location”  and terminology like “superb performance” need wording. 

Round 2

Reviewer 2 Report

Comments and Suggestions for Authors

The authors have satisfactorily addressed all my comments.

Reviewer 3 Report

Comments and Suggestions for Authors

The manuscript seems publishable now.